# HOX-Gene Cluster Organization and Genome Duplications in Fishes and Mammals: Transcript Variant Distribution along the Anterior–Posterior Axis

**DOI:** 10.3390/ijms23179990

**Published:** 2022-09-01

**Authors:** Nikolay Ozernyuk, Dimitry Schepetov

**Affiliations:** 1Koltzov Institute of Developmental Biology of Russian Academy of Sciences, 26 Vavilov Street, 119334 Moscow, Russia; 2Faculty of Biology, Lomonosov Moscow State University, 1-12 Leninskie Gory, 119991 Moscow, Russia

**Keywords:** HOX genes, transcript isoforms, anterior-posterior axis, co-optations, body plan

## Abstract

Hox genes play a crucial role in morphogenesis, especially in anterior–posterior body axis patterning. The organization of Hox clusters in vertebrates is a result of several genome duplications: two rounds of duplication in the ancestors of all vertebrates and a third round that was specific for teleost fishes. Teleostei cluster structure has been significantly modified in the evolutionary processes by Hox gene losses and co-options, while mammals show no such tendency. In mammals, the Hox gene number in a single cluster is stable and generally large, and the numbers are similar to those in the Chondrichthyes. Hox gene alternative splicing activity slightly differs between fishes and mammals. Fishes and mammals have differences in their known alternative splicing activity for Hox gene distribution along the anterior–posterior body axis. The analyzed fish groups—the Coelacanthiformes, Chondrichthyes, and Teleostei—all have higher known alternative mRNA numbers from the anterior and posterior regions, whereas mammals have a more uniform Hox transcript distribution along this axis. In fishes, most Hox transcripts produce functioning proteins, whereas mammals have significantly more known transcripts that do not produce functioning proteins.

## 1. Introduction

Vertebrate genomes have been significantly impacted by polyploidization. As a result, the number and structure of Hox clusters in different vertebrates is remarkably variable. In two rounds of complete genome duplication in vertebrates, except for teleost fishes, four Hox clusters were formed from one primary cluster in the early evolutionary history of the group [1,2,3,4,5]. The first Hox cluster duplication occurred in the common ancestor of the Agnata and Gnathostomata fishes about 500 million years ago (Mya) [6]. Subsequently, Gnathostomata underwent a second genome duplication that led to the formation of four Hox clusters (abbreviated as 2R-WGD).

An additional third round of fish-specific genome duplication (abbreviated as 3R-WGD) occurred at about 350 Mya; this led to large-scale evolutionary radiation and the formation of a new group—teleost fishes (Teleostei), which number more than 20,000 species [7,8,9,10]. As a result, the number of *Hox* gene clusters in teleost fishes increased from four to eight. The third genome duplication round did not affect the older fish groups of Sarcopterygii and Chondrichthyes. An ancient branch of the lobe-finned fish Sarcopterygii—the Coelacanthiformes group, which includes the coelacanth, *Latimeria chalumnae*—has only four Hox clusters. A similar picture is also typical of the Elasmobranchii, which includes sharks, skates, and chimeras. The feline shark, *Scyliorhinus canicular*, has four Hox clusters. Thus, there are two main *Hox* cluster architecture types in fishes: the 2R-WGD cluster type and the 3R-WGD teleost type [2,10]. One more local Hox gene duplication round occurred around 25–100 Mya in salmon fishes [11,12,13]. The contemporary Atlantic salmon, *Salmo salar*, and the rainbow trout, *Oncorhynchus mykiss* have 13 Hox clusters that contain 118 Hox genes, including eight pseudogenes. This is the largest known number of genes present in fishes [14].

Animal HOX-gene transcripts go through the usual stages of RNA processing: alternative splicing, alternative promoter use, and alternative polyadenylation. Alternative splicing has been reported to play a significant role in differential transcript formation in terms of sex determination, receptor synthesis, morphological differentiation (i.e., different muscle types), and Hox gene functioning [15]. Splicing is the most versatile regulation mechanism; other processing stages augment its regulative abilities [3,16,17]. Morphogenetic complexity in different embryo regions varies greatly along the anterior–posterior axis and can be linked with alternative splicing activity, as higher patterning complexity requires more Hox protein variants.

Some genes are producing more than one protein via alternative splicing, giving an estimated at least ~20 fold increase of proteins to coding genes in mammals [18,19,20]. Relative impact of alternative splicing on produced proteins rises from *Сaenorhabditis elegans* to human 5–6 times, according to different estimation approaches [21].

Hox gene organization and their alternative splicing were analyzed in several fish (zebrafish, fugu, pufferfish, Japanese medaka, tilapia, platyfish, and latimeria) and mammal species (human, mouse, bonobo, elephant, cat, dolphin, and opossum). We studied the Hox gene organization features in fishes and mammals, which differ in the Hox cluster number caused by the differences in the numbers of genome duplication rounds in these vertebrate groups. The Hox gene numbers, their cluster structure, and alternative splicing activity were analyzed in various fish groups; Coelacanthiformes, different evolutionary branches of the Teleostei, and mammals have various Hox gene numbers and are distinguished by the cluster organization of these genes. We used open, verified, and annotated data on the transcriptomes of the selected species. The latest assemblies from the Ensembl genome database were used, accessed via the Biomart tool. Specific assemblies and gene accession nubmers are listed in Appendix A. The Hox gene transcript and exon numbers for the selected organisms were used. For statistics and visualization purposes, we used the R program, run in an R-Studio environment.

## 2. Cluster Organization of Vertebrate Hox Genes

The *Hox* cluster structure in Bilateria is related to their axial regionalization function. In *Hox* genes, the cluster organization and the mechanisms determining the anterior–posterior body plan are associated with the spatial arrangement order of these genes in the cluster [22,23,24]. These cluster structures are different in invertebrates and vertebrates. Vertebrates demonstrate compactly organized Hox clusters. The presence of several clusters in vertebrates makes it possible to pattern morphogenetic processes in the anterior–posterior and dorsoventral body axes. The Hox cluster organization in vertebrates is related to several genome duplication rounds and impacts the subsequent clusters and their evolutionary changes.

### 2.1. Genome Duplications in Vertebrates: Hox Genes Loss and Co-Options

The fish cluster structure was significantly modified by Hox genes losses and co-options. The Hox gene loss and co-option in different lineages reflect their evolutionary history. In teleost fishes, specific genome duplications led to a noticeable loss of *Hox* genes and even of whole clusters [2,16,25,26]. Consecutive Hox genes and whole-cluster loss are, probably, a result of their excess numbers after duplication. These significant gene losses are common to almost all fishes, not only in Teleostei [16,27,28] but also in some elasmobranch fishes (sharks and rays) [29] and coelacanths [16]. In the beginning, it was assumed that these losses in Teleostei fishes were the result of Hox gene redundancy after 3R-WGD genome duplication, but Hox gene losses in elasmobranch fishes and coelacanths may also be due to other causes.

Especially noticeable were the Hox cluster evolutionary secondary loss events in fish following 3R-WGD genome duplications. Many fish species, including pufferfish [30], stickleback [31], medaka [32,33] and tilapia [34,35] lost the *HoxCb* cluster. In other fish species, such as zebrafish (*Danio rerio*) [25] and blunt snout bream [36], the *HoxDb* cluster is also lost.

Some fishes that did not pass 3R-WGD genome duplication, such as the elasmobranch fishes—feline sharks (*Scyliorhinus canicular*) and rhomboid stingrays (*Leucoraja erinacea*) also lost the *HoxС*-cluster [16,29]. Nevertheless, some Teleostei retained the complete Hox cluster set, as seen in the European eel, *Anguilla anguilla* [37].

A special example of a change in Hox gene numbers and their clusters is found in sturgeons [38]. The Acipenseridae went through a local segment rediploidization at approximately 200 Mya. As a result, the sterlet (*Acipenser ruthenus*) has eight Hox clusters, containing 87 Hox genes and one pseudo-gene (Hox-14). The sterlet retained the most intact set of 3R Hox clusters and the highest number of 3R Hox genes. This group has notoriously low evolutionary clock rates [38]. Thus, the Hox genes and their cluster organization in different fish groups vary significantly.

### 2.2. Hox Clusters Structure

The main evolutionary trend of Hox clusters after genome duplication is gene loss [39]. Vertebrate *Hox* clusters have no transposons and are extremely dense, whereas in invertebrates, the cluster is much less compact [26,40,41,42]. The differences in Hox cluster compactness is due to the intergenic distance reduction that influences the *cis*-regulatory elements. Short intergenic regions in vertebrates could indicate simpler *cis*-regulatory elements appearing with a large number of Hox genes [26,28].

Compactly organized Hox clusters in vertebrates are mostly transcribed in one direction. This is significantly different from the clusters seen in invertebrates, which have disorganized (Сaenorhabditis, Echinodermata), split (Ciona), and atomized clusters (Oikopleura) [26,43]. The organization of Hox clusters in fish and mammals has a different evolutionary history associated with several genome duplication rounds. Fishes have a tendency for evolutionary change in the Hox clusters and the gene numbers in single clusters. This process is associated with evolutionary Hox gene loss and co-option. Analyzed mammals have compact and more efficiently organized Hox clusters.

The cluster structures in the analyzed fish show different Hox-gene numbers in individual clusters from various fish species: Chondrostei (*Acipenser ruthenus*—11.0), the ancient relict group of Coelacanthiformes (*Latimeria chalumnae*—6.5), the Teleostei (Salmoniformes (*Salmo salar* and *Oncorhynchus mykiss*)—8.5). It is significant to note that the lowest Hox gene numbers in a single cluster can be found in the Teleostei: *Danio*, *Oryzias*, *Takifugu*, *Tetraodon*, *Gasterosteus*, and *Xiphophorus*, which have an average number of 5.1 genes per cluster (Table 1). At the same time, these species have the largest known cluster numbers. These effects may be related to Hox gene redundancy in teleost fishes after 3R-WGD genome duplication.

Another fish Нoх-cluster feature is their size, which is influenced by the size of the intergenic region. In the Teleostei, the zebrafish has НoхA cluster paralogs sized ~58 kb for HoxAa and ~33 kb for НoхАb [44]. Similar size ranges of these clusters are found in fugu and tilapia [35,45]. In mammals, the Hox genes and the organization of their clusters are different from those of fishes [46,47,48,49]. Mammalian Нoх clusters are 100–120 kb long, which is much larger than in Teleostei fishes [28]. Invertebrate Hox clusters are drastically larger, with a typical size of over 1000 kb [40]. In the case of mammals and elasmobranch fishes that have only 2R-WGD genome duplication, these sizes coincide. Thus, the shorter intergene spacers of vertebrate clusters are simpler *cis*-regulatory elements affecting more target Hox genes [28].

## 3. Hox Genes and Alternative Splicing

Alternative splicing significantly boosts the protein variants produced by some genes. It is estimated that 20,000 genes encode 0.4–1 million different proteins in mammals [18]. This makes alternative splicing a powerful mechanism that is able to promote rapid evolutionary change. This process notably impacts morphogenetic specification [18,19,20]. In the *Caenorhabditis elegans* nematode, alternative splicing is known in 15% of its genes, in *Drosophila*, it is ~40%, while for humans it is reported to be from 75 to 94%, based on different estimation approaches [21]. Alternative splicing has been reported to play a significant role in differential transcript formation in sex determination, receptor synthesis, tissue differentiation (i.e., different muscle types), and Hox gene functioning [3,15,21].

Animal Hox-gene transcripts go through the usual stages of RNA processing: alternative splicing, alternative promoter use, and alternative polyadenylation. Splicing is the most versatile regulation mechanism and other processing stages augment its regulative abilities [3,16,17]. Morphogenetic complexity in different embryo areas varies greatly along the anterior–posterior axis and can be linked with alternative splicing activity as higher patterning complexity requires more Hox protein variants. To estimate the possible role of alternative splicing in Hox gene functioning, we compared the analyzed fishes and mammals with known transcripts, covering a variety of taxons, with different Hox genes and their cluster numbers. Aside from some forms of regulatory activity, there may be other possible reasons for the increase in alternative isoforms in HOX gene transcripts. However, if there is indeed some indirect reason for the elevated activity of alternative splicing in certain regions along the anterior–posterior axis, this is still an indication of more complex regulatory activity.

Dedicated Hox-gene alternative splicing activity studies are scarce, especially in fishes. In particular, the *Scyliorhinus canicula* shark is shown to have the *HoxA11*, *HoxB9*, *HoxB4*, *HoxB3*, and *HoxD3* genes [16]. In mammals, *Hoxb-3* and *Hoxa-5* were shown to produce various mRNA in mice [50,51]. Several transcripts obtained via alternative splicing and promotors were also shown for mouse НoхА9 [52,53]. In humans, several mRNA types are known for *HoxC-6*, *HoxC-5*, *HoxC-4* [54], and *НoхА-7* [27].

However, there is a mass of data suitable for Hox gene splicing form abundance analysis. Mass-throughput sequencing technology gave us an opportunity to use the existing generalized data for vertebrate genomes and transcriptomes. We used such data to compare the numbers of known alternative mRNA for all Hox genes in fish and mammal species. We have summarized the current information on Hox gene features and their cluster organization and have analyzed the data on their alternative splicing in various groups of the Coelacanthiformes, different Teleostei evolutionary branches, and mammals in Table 1. It should be noted that most transcript sets are, obviously, incomplete; in some cases, there are no known products for some genes found in the genome. Nevertheless, we believe that even incomplete data is indicative of relative transcript variability, as random subsets from a complete transcript isoform set should retain their disproportionality if they exist. Thus, our main focus is on the relative inequality of splice forms and not on their exact quantities.

Fishes and mammals show differences in known alternative splicing activity for HOX genes (Table 1). This feature is characteristic both for each Hox cluster and for the summarized data for all Hox clusters in individual species. Not all genes (corresponding to the expected numbers for existing clusters) are known to actively produce the products of fish Hox clusters. *Danio* has all 48 Нoх-genes that produce mRNAs, while *Tetraodon* has only 70%, and for *Takifugu*, only 38% are known to produce transcripts. The mean number of HOX-gene alternative transcripts per gene also differs among fish species (Table 1). The data for mammals differ significantly in comparison. All analyzed species have all expected genes active, with different alternative splicing activity patterns. However, the overall levels of alternative splicing in mammals and fishes are strikingly similar.

Hox gene expression in vertebrates along the anterior–posterior axis can be divided into several regions (anterior, central, and posterior) with limits being different for the different classes [55,56]. Notably, the expression of neighboring Нoх genes, functioning as selector genes to pattern the various morphogenetic processes, can overlap and change [46,49,57,58]. Hox gene expression in the anterior region is especially interesting, as many important and complex morphogenetic processes occur in the head’s main structure development: craniofacial morphogenesis, derived from the cranial neural crest cells and hindbrain. An essential role in this region’s morphogenetic patterning is played by the Hox3 genes, located in the cervical area between the anterior and central regions. Fishes and mammals show differences in the distribution of Hox-gene expression along the anterior–posterior axis.

Several Hox clusters present in vertebrates ensure morphogenetic process control in a dorsoventral direction. Comparison of the alternative splicing activity of the same Hox genes in each cluster revealed differences in the analyzed fishes and mammals along the dorsoventral direction. However, no regularities were found.

### 3.1. Fishes

Hox gene alternative splicing activity for several fish species from different evolutionary lines varies both between species and along the anterior–posterior axis (Figure 1). The analyzed fish groups of the Coelacanthiformes and Teleostei have higher numbers of known alternative spliced mRNA from the anterior and posterior regions.

The fish anterior region specified by *Нoх-1*, *Нoх-2*, and also *Нoх-3* occupies an intermediate position between the anterior and central regions. The latter is considered intermediate as it is next to the central region *Нoх-4* gene. Genes of this region specify several types of complex morphogenesis formation: the cranial neural crest and hindbrain, consisting of a rhombomere series [16,57,58]. In zebrafish embryos, the regional hindbrain differentiation is coordinated by: *Hoxb-1*, *Hoxa-2*, *Hoxb-2*, *Hoxb-3*, and *Hoxd-3* [58], co-expressing in rhombomeres r2 to r6. Paralogous Нoх-2 in fish embryos takes part in pharyngeal arch and hindbrain regional differentiation; however, this is to a lesser extent than *Нoх-3* [57]. The second pharyngeal arch differentiation is regulated by a combination of *Нoх2* paralogs that vary in the different fish species [57]. The *Нoх-3* gene takes part in the regional differentiation of hindbrain structures, anterior somites, and their derivatives. This Hox gene is characterized by the greatest level of alternative splicing activity in most analyzed fish species (Figure 1). It demonstrates an obvious overlap in function with the neighboring *Нoх-2* and *Нoх-4*. Both *Нoх-3* and *Нoх-4* play a role in anterior somite differentiation. Thus, in the anterior region, several morphogenetic processes that are crucial to normal embryonic development take place. This complexity and diversity of structures can be the reason for the elevated alternative splicing activity in this region.

In the central region, *Нoх-4*–*Нoх-8* genes take part in the regional differentiation of the hindbrain (together with anterior region genes), somitogenesis, and the later differentiation of the axial skeleton, including the ribs [57,58]. In zebrafish embryos, hindbrain regional differentiation is guided by *Hoxb-4* and the anterior *Нoх* genes [58]. The paralogs *Нoхb-4* and *Hoxd-4* are expressed at the r6/r7 rhombomere borders. In the central region, far fewer alternative transcripts are known for the Hox genes (Figure 1), which can be explained by the relative somite structure simplicity and, therefore, less complex processes guiding their formation.

In the posterior region (*Нoх-9*–*Нoх-13*), fin specification occurs and the posterior part of the somites is patterned. Pectoral and pelvic fin development is rather complex and is guided by posterior region Hox genes, in the same way as the hind legs in tetrapods [57,58,59,60,61]. A unique perspective on this change in fin development, leading to terrestrial vertebrate limbs, can be given by the paleontological data available on ancient fish species, such as *Panderichthus* [62], *Ventastega* [60], *Tiktaalik* [63], and ancient amphibians [59,60,64]. The Нoх genes of the posterior region were co-opted in the genome in early terrestrial vertebrate evolution and later underwent tandem duplications [28,63]. This probably happened in all tetrapod common ancestors, at some time between the advent of *Panderichthus* and *Acanthostega* [59,60].

The contemporary ray-finned fish *Polyodon* shows regulatory elements in fin development controlling *HoxA* cluster transcription, including the most distal fin parts [61]. This process in fishes is remarkably similar to that of distal limb development in tetrapods. However, fishes lack the 5′ enhancer at the end of the НoхА cluster that regulates finger formation in tetrapods [65]. Thus, in the fish posterior region, *Нoх*-genes are involved in the fins’ regional specification and the linked complex morphogenetic processes. The major event in fin development is mass cell migration to the distal parts of the growing fins. The development of both distal and proximal parts in the pectoral and pelvic fins is patterned by the interaction of the *НoхА-11*, *НoхА-13*, and *HoxD-13* genes [59]. Thus, these complex processes may be the reason for the elevated alternative Hox gene-splicing activity in the fish posterior region.

#### The Distribution in the Hox Genes of Paralogous Clusters after Three Genome Duplications

The fish species that have undergone three rounds of gene duplication serve as a conventional model for analyzing the Hox gene splice-form distribution between paralogous copies. We have shown that the Hox gene splice-form distribution in paralogous copies in the Teleostei follows a certain asymmetric pattern. In particular, in zebrafish Hox clusters, the number of splice forms in more populated initial clusters (Aa, Ba, Ca) is notably higher compared to the number in their paralogous copies from the last duplication (Ab, Bb, and Cb) (Table 2).

Similar differences in the Hox splice-form distribution in paralogous copies, to a lesser extent, can be seen for the other fish species. Notably, in the more populated clusters, only part of the Hox genes produce the splice forms, while in the paralogous copies of the Hox genes, almost all have alternative splice forms. The differences between these copies can be related to asymmetric Hox gene loss after their fish-specific duplication. Thus, differences between the paralogous clusters’ A, B and C copies can be explained by the different evolutionary fates of their paralogs.

## 4. Mammals

The Hox cluster parameters in fishes and mammals have some functional peculiarities. We compared the Hox genes’ alternative splicing activity distribution along the anterior–posterior body axis in these vertebrate groups. In particular, unlike fishes, mammals have uniform alternative transcript distribution along the Hox clusters.

### Hox Gene Alternative Splicing Activity along the Anterior–Posterior Axis

Interspecific differences in mammals’ Hox gene alternative splicing activity are similar to the analyzed fishes. In mice and humans, the known Hox gene transcript number is very similar, as is their distribution along the anterior–posterior body axis. The mammalian Hox cluster is divided into the cervical (*Нoх1*–*Нoх5*), thoracal (*Нoх6*–*Нoх8*), and abdominal (*Нoх-9*–*Нoх-13*) regions, the latter including the lumbar, sacral, and tail parts [55]. In the cervical region, *Нoх**-2*, *Нoх-3*, and *Нoх-4* and their splice forms are most prominent. In particular, *Нoх-3* plays a crucial role in neck regional specification and has the highest known splice-form number (Figure 1). The cervical region Hox genes pattern such structures as the neural crest, hindbrain, the spinal cord’s upper parts, some of the vertebrae, and the ribs. The specification of these processes is, to a certain extent, similar to that in fish development. Mouse *Нoха-2* and *Нoха-3* define the development of the neural crest and its derivatives—neck cartilage and the thyroid gland [46,47,66].

Neck Hox-gene paralogs were shown to play an important role in certain regional specifications [47]. Mice that are homozygous in *Нoха-3* mutations have an abnormality in pharyngeal arch development. The *Нoха-3* and *Нoхd-3* paralogs perform detailed neural crest regionalization in the dorsoventral axis. These genes act synergistically in structure patterning. *Hoxa-3* knockout leads to neural crest development defects, while *Hoxd3* knockout leads to the disruption of the development of С1 and С2 somites and the corresponding vertebrae. A dysfunctional *Hoxd-3* leads to neck vertebra development anomalies. If both *Hoxa-3* and *Hoxd-3* are disrupted, the atlas vertebra does not form during development [47]. *Нoха-3* homozygous knockout causes damage to neural crest cells, leading to neck cartilage deficiency, heart and blood vessel anomalies, and the absence or malformation of the thymus, thyroid, and parathyroid glands.

*Нoх-4* in mice is expressed in the presumptive hindbrain and upper spinal cord, somites, and the derivative prevertebrae [67]. Homozygous mutations in this gene may cause abnormal neck, vertebra, and rib development. Mutant newborn mice have ribs on the C7 vertebra. Transgenic mice with *Hox-4* overexpression showed suppressed rib development of the C7 ribs [67]. *Нoха-5* plays a crucial role in mammal neck morphogenesis [49]. It is involved in skeletal structure regional differentiation. Mice with a mutant Нoха5^−1^ have severe axial skeleton development distortions, including gomeotic transformation and breastbone and respiratory system malformation. Mice do not have a distinct border between the neck and the cervical regions [49].

*Нoх* gene expression during mammalian somite formation has several specific features [48]. During somite differentiation, starting with the induced paraxial mesoderm and until complete axial skeleton patterning, Hox gene expression can shift along the anterior–posterior axis; later in development, this ceases to occur. In the presomite mesoderm, these gene patterns somite axially. It was proved experimentally with *Нoха-10* overexpression. Thus, a significant complexity of morphogenetic processes in the cervical and thoracic regions correlates with high Hox-gene alternative splicing activity along the anterior–posterior axis.

For mammals, similar to fishes, the posterior (*Нoх-9*–*Нoх-13*) regional identification of morphogenesis is majorly guided by the A and D clusters of these genes [46]. The comparison of limb morphogenesis with fin development reveals a significant similarity, despite obvious differences in their complexity. Hox gene mutations and knockouts give much information on limb regionalization in mammals. In an analysis of *Hoxa11*/*Hoxd11* and in *Hoxa11*/*Hoxd12* mice, the doubled mutants indicate that these genes patterned limb development, with radius and ulna formation during the specification prechondrogenic condensation and during the growth of these bones [46]. The disruption of *Нoха-11* and *Нoхd-11* results in *radius* and *ulna* loss. Patients with homozygous mutations in *Hoxd-13* are registered to have hand and feet development anomalies, known as syndactyly. Patients with homozygous HoxA-13 mutations are known to have distal limb development anomalies [68]. In the mammal embryo posterior cluster region, the Hox genes pattern shows different morphogenesis for the fore- and hindlimbs. Significantly, targeted mutations in Hoxa-9 and Hoxb-9 revealed synergistic interactions [69]. Thus, mammalian morphogenesis is characterized by a higher interconnection level on the anatomical and regulatory levels that are guided by Hox genes. Thus, the levels of Hox gene alternative splicing in mammals show no significant differences along the anterior–posterior body axis, which may be explained by their morphogenetic peculiarities. These differences from fish development are, probably, a result of the different evolutionary histories in these two groups, mainly the number of genome duplication rounds. The representatives of Afrotheria, Euarchontoglires, Laurasiatheria, and Marsupialia show no obvious differences. It should be noted that the observed differences and similarities between mammalian lineages are not due only to their phylogenetic relationship. Available data is partially incomplete and species that are studied better (i.e., human and mouse) have more coverage, thus potentially explaining the observed differences between closely related species (e.g., humans and bonobos).

## 5. Protein-Producing Hox Transcripts

Hox gene transcript analysis showed that not all mRNAs in alternative splicing were the matrix for functioning properties synthesis. In the analyzed fish species, most Hox gene transcripts produce viable proteins: they have no open reading frame, have a retained intron, or are otherwise predicted to be unfit for proper protein synthesis. Only the fractions of certain transcripts are too short and have no homeobox domain, or are otherwise not viable for functioning protein production. In zebrafish, Нoх-3 paralogs (*Нoхb3*, *Hoxc3*, and *Hoxd3*) are known to produce 17 transcripts and 15 coding functioning proteins (Figure 2). A similar transcript structure was found for the Hox4 gene. In this fish species, the Hox7 paralog *Hoxb7* is known to produce 4 transcripts but only 2 coding functioning proteins. In other fish species (fugu, tatraodon, and tilapia), all the known Hox transcripts code functional proteins; however, this may be as a result of data deficiency.

Surprisingly, the number of known Hox transcripts in mammals that code proteins is significantly greater than in fishes. In humans, only three Hox genes do not produce non-coding mRNAs and, in mice, only two such Hox genes are known. Overall, the abundance of known non-protein-coding transcripts in the human transcriptome is drastically higher (Figure 2). In other, less well-studied mammalian species, no non-coding Hox gene mRNAs are known; however, similarly to fishes, this may be due to insufficient data. The cause for sufficient differences in the Hox transcripts between analyzed fishes and mammals is not clear. It is possible that these mRNAs have a regulatory function similar to long non-coding RNA.

Concerning the mechanism of non-coding Hox gene transcript emergence, it should be noted that during the alternative splicing process, errors can occur, particularly due to exon deletion. No function has ever been shown for non-coding Hox gene transcripts. It has been shown that long non-coding RNAs, in general, are known to perform certain regulatory functions on chromatin packaging levels [3,70]. The long non-coding RNAs can interact with transcription factors. It is possible that non-coding Hox gene transcripts may be involved in the regulation of these genes’ functions. However, other explanations are also possible, and a separate series of experimental studies are needed to verify whether non-protein-coding transcripts of the HOX genes are a side product of some other process, or whether they perform a certain function on their own. The same is true for this hypothetical function.

## 6. Changes in Hox Cluster Organization and Transcription in Different Lineages

Differences and similarities within and between mammals and fishes should be considered in the context of vertebrate evolution and phylogenetic relationships. Not all vertebrate groups are equally well-studied in terms of Hox cluster organization; only model organisms (i.e., zebrafish, human, and mouse models) have good coverage in transcript variation. Nevertheless, we tried to sample from better-covered (having at least some known isoforms) transcriptomes for both fishes and mammals. Both representatives of the Actinopterygii and Sarcopterygii have somewhat elevated known alternative spliced mRNA numbers from the anterior and posterior regions, despite their different numbers of Hox clusters, due to the different numbers of whole-genome duplication rounds. In this respect, the Coelacanthiformes should be expected to be quite similar to tetrapods. There are no cluster losses in Latimeria, setting it aside from bony fishes, and this is in agreement with the known rigidity of vertebrate Hox cluster organization [28,71]. However, the coelacanth has an elevated number of transcript isoforms in the neck region, which is similar to that of other studied fishes. This may be connected to a similarity in their body plans, as was discussed above. It should be of considerable interest to compare both the Coelacanthiformes and Teleostei to Chondrostei and Chondrichthyes. However, no complete data are available as of today for sturgeons, while the shark data [16] are not directly compatible with Ensembl. Nevertheless, the published data for *Scyliorhinus canicula* show transcript distribution along the anterior–posterior axis with patterns similar to other fishes. Another significant point of interest is those fish species that had more whole-genome duplication rounds than Teleostei—the Acipenseriformes and Salmoniformes. Unfortunately, salmon transcriptomes are yet not fully annotated, and sturgeon data are unavailable. Both groups would be challenging to investigate, as it is technically difficult to assemble genomes and transcriptomes in polyploid species that have had recent duplications.

In mammals, divergence times and, consequently, genome dissimilarities between lineages are notably smaller than between Actinopterygii and Sarcopterygii. The main differences between the mammalian and fish Hox clusters are, probably, as a result of their different evolutionary histories, mainly the number of genome duplication rounds. Representatives of the Afrotheria, Euarchontoglires, Laurasiatheria, and Marsupialia families are all similar, as far as can be seen from the known transcripts. It should be noted that the observed differences and similarities between mammalian lineages are not solely due to their phylogenetic relationship. The available data are partially incomplete and species that are studied more fully (i.e., human and mouse) have more coverage, thus potentially explaining the observed differences between closely related species (e.g., human and bonobo). Such conformity between mammals is not surprising, as they had no additional genome duplications and, thus, should be subject to vertebrate constraints in Hox cluster rearrangement [28]. Mammals are also a relatively young group, even among the tetrapods. The perceived differences between mammal lineages are greater than between fishes, mostly due to a bias. The precise details of fish anatomy, development, and physiology are far less widely studied. The addition of data on other tetrapods, especially amphibians, would further enhance our understanding of Hox cluster evolution in vertebrates in the context of genome duplication rounds and transcript isoform variation along the anterior–posterior axis.

## 7. Conclusions

Hox-gene cluster organization is key to their role as they provide developmental regionalization in the anterior–posterior and dorsoventral body axes. This makes it important to compare cluster organization in diverse vertebrate groups with different morphogenesis and evolutionary histories. The number and structure of Hox clusters in different vertebrates are remarkably variable. In these animals, the Hox genes and their cluster organization depend on the number of evolutionary genome duplications. These animals have gone through at least two rounds of genome duplications. That has led to four Hox cluster formations, while a third specific genome duplication in teleost fishes resulted in a Hox cluster increase from four to eight. As a result, Hox gene structure and their cluster organization in fishes and mammals are different. In the Teleostei, the third genome duplication round led to a certain level of Hox gene evolutionary redundancy, and to their subsequent losses and co-options. This, in turn, led to different gene numbers in the various fish groups. The analyzed mammals have constant Hox gene numbers in each cluster and no tendency to evolutionary loss and co-option. Fish show a tendency to evolutionary changes in Hox gene numbers in a wide range for a single cluster, whereas mammals have constant members of genes in each cluster. These differences are probably associated with the particular features of morphogenetic processes in fishes and mammals, firstly in terms of their complexity. Differences between these vertebrate groups are also associated with their evolutionary history. A unique feature of fishes is that the third fish-specific genomic duplication led to large-scale evolutionary radiation and the rise of a new fish group—the Teleostei.

Vertebrates are characterized by compactly organized Hox clusters having a common transcriptional direction. However, noticeable differences between the Teleostei, other fish groups (Coelacanthiformes and Elasmobranchii), and mammals are also evident in Hox gene cluster organization. Fishes have a tendency toward changes in the Hox clusters and gene numbers in single clusters that are associated with their evolutionary loss and co-option. Mammalian Hox gene clusters have compact and efficient organization.

Various vertebrates’ body region specification is linked to morphogenetic process variability, which is guided by a limited number of Hox genes. Hox gene transcripts go through the usual stages of RNA processing: alternative splicing, alternative polyadenylation, and alternative promoter use. The alternative splicing of these genes is the main mechanism to cope with this disproportionality. The overall alternative splicing activity in fish and mammals differs slightly, but this activity distribution along the anterior–posterior body axis and the protein-coding and non-coding transcript ratios differ remarkably. The analyzed fish species all have higher numbers of known alternative mRNA from the anterior and posterior regions, whereas mammals have a uniform Hox gene alternative transcript distribution along the anterior–posterior body axis. These differences are, probably, associated with the morphogenetic features of the early ontogenesis of fish and mammals. Notably, the ratio of protein-coding and non-coding Hox transcripts in fishes and mammals sharply differ. In fishes, the non-coding Hox transcript fraction is negligible, while in mammals, these transcripts are significantly greater.

It should be noted that all that is discussed above is based on open data resulting from studies that are not explicitly focused on the alternative transcripts of HOX genes. Assumptions of the role of transcript isoforms and the hypothetical role of non-coding transcripts are purely theoretical and, on their own, only highlight a potential point of interest for further dedicated experimental research.

## Figures and Tables

**Figure 1 ijms-23-09990-f001:**
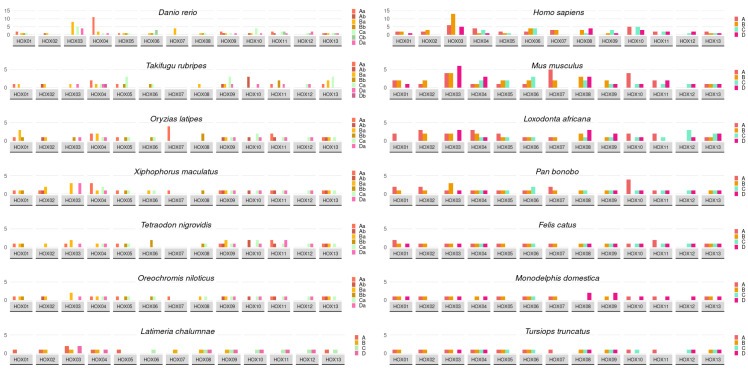
Known transcript numbers along the anterior–posterior axis. Clusters are coded by color, as per right-side legend.

**Figure 2 ijms-23-09990-f002:**
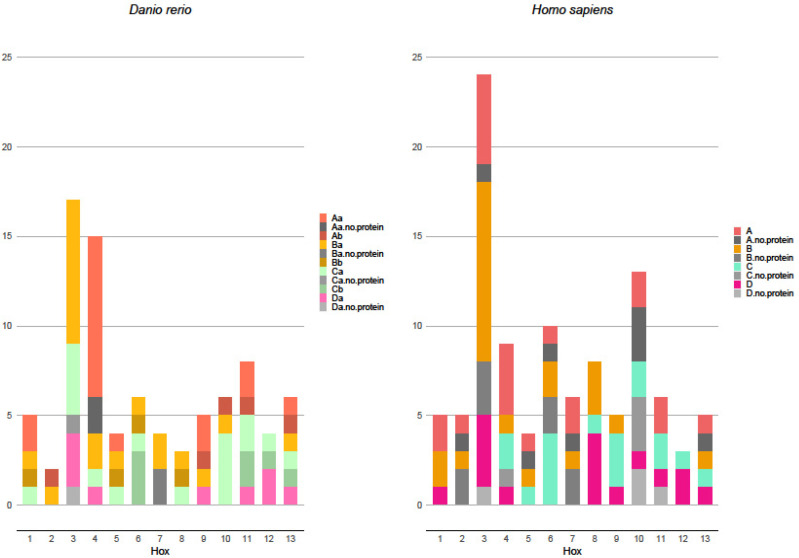
Zebrafish and human transcript numbers along the anterior–posterior axis, with marked non-viable protein-coding transcripts.

**Table 1 ijms-23-09990-t001:** Known active HOX genes and their transcript abundance.

Species	Clusters with Known Transcripts	Genes with Known Transcripts	Genes with Alternative Transcripts	Total Known Transcripts	Average Transcript per Active Gene	Used Transcriptome Assembly
*Danio rerio*	7	48	14	87	1.81	GRCz11
*Takifugu rubripes*	7	31	7	42	1.35	fTakRub1.2
*Oryzias latipes*	6	35	7	45	1.29	ASM223467v1
*Xiphophorus maculatus*	6	37	5	45	1.22	X_maculatus-5.0-male
*Tetraodon nigroviridis*	6	34	7	41	1.21	TETRAODON 8.0
*Oreochromis niloticus*	6	36	1	37	1.03	O_niloticus_UMD_NMBU
*Latimeria chalumnae*	4	26	2	28	1.08	LatCha1
*Homo sapiens*	4	38	26	103	2.71	GRCh38.p13
*Mus musculus*	4	38	21	76	2.00	GRCm39
*Loxodonta africana*	4	35	17	57	1.63	Loxafr3.0
*Pan bonobo*	4	35	6	44	1.26	panpan1.1
*Felis catus*	4	37	2	39	1.05	Felis_catus_9.0
*Monodelphis domestica*	4	28	2	30	1.07	ASM229v1
*Tursiops truncatus*	4	32	0	32	1.00	turTru1

**Table 2 ijms-23-09990-t002:** The distribution of Hox genes and the activity of their alternative splicing in the initial (Aa, Ba, Ca, Da) and alternative copies (Ab, Bb, Cb) in teleost fishes. The number of active genes is shown without brackets, and the number of genes with alternative transcripts is in brackets.

Species	Aa	Ab	Ba	Bb	Ca	Cb	Da
*Danio rerio*	6 (4)	5 (0)	11 (3)	4 (0)	11 (3)	4 (2)	7 (2)
*Tetraodon nigroviridis*	6 (1)	4 (1)	6 (2)	4 (1)	8 (1)	-	6 (1)
*Takifugu rubripes*	5 (1)	3 (1)	7 (1)	3 (1)	6 (3)	-	6 (0)
*Oreochromis niloticus*	6 (4)	5 (0)	7 (3)	3 (0)	9 (3)	-	6 (2)
*Oryzias latipes*	6 (3)	5 (0)	5 (2)	5 (1)	8 (1)	-	6 (0)
*Xiphophorus maculatus*	8 (1)	5 (0)	7 (2)	3 (0)	8 (1)	-	6 (1)

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
