# Peer review of "HOX-Gene Cluster Organization and Genome Duplications in Fishes and Mammals: Transcript Variant Distribution along the Anterior–Posterior Axis"

_ijms, 2022, doi:10.3390/ijms23179990_

Round 1

Reviewer 1 Report

The review is, in fact, much more about the distribution of Isoforms/Splice variants in the Hox cluster than the "HOX-Gene cluster organization and genome duplications". Also, in terms of  number of Hox genes and clusters etc. there is  little update beyond the review Pacual-Anaya et al.

I therefore suggest to sharpen the focus even further on the isoforms and to change the title to reflect this focus.

I found the panels in Fig.1 hard to read. It would seem better to separate the A-D clusters into four parallel lines and to show the # of isoforms for each of them, instead of the combined bar. I'm not sure if separating the teleost a/b clusters as well is helpful and shows much in addition.

To what extent do the authors believe that the reported isoforms are complete sets? The comparison of Homo sapiens and Pan bonobo, if fact suggests that there is a strong ascertainment bias and a large number of isoforms is likely of have gone undetected in most species [Human Mouse and Zebtrafish showing a much larger number would suggest that simply more distinct transcripts have been reported in better-studied species. While it makes sense to review what has actually been reported, the quantitiative comparisons do serve a bit more discussion with regard to such biases.

Author Response

Dear Reviewer,

Thank you, for your input!

We addressed your suggestions as follows:

Title was changed to "HOX-Gene cluster organization and genome duplications in fishes and mammals: transcript variant distribution along anterior-posterior axis" to better represent actual manuscript contents, and discussion on the subject was expanded.

We modified Fig.1 according to your suggestion.

We do not believe most available datasets to include complete isoform sets, however it should not impact their abundance ratio. We expanded our manuscript to further clarify and discuss this issue.

Reviewer 2 Report

In this review, the authors first summarize vertebrate Hox gene evolution with a focus on Hox cluster organization, including duplications and losses in fish species. This part, however, is not a topical review, as the newest publications that are cited are from 2015 and 2016, both of which are no research papers either. A somewhat novel topic emerges once the authors propose ideas to explain the seemingly abundance of alternative Hox gene transcripts in certain Hox genes and their lack in others. From the databases, the authors identified alternative transcripts of Hox genes and argue that certain Hox numbers preferentially seem to have evolved alternative transcripts.  The authors speculate that alternative transcripts, from splicing events or alternative promoters, may increase the levels of regulations between Hox genes and go on to suggest that these transcripts are more abundant for Hox genes that are from regions along the body axis where the coordination of developmental programs seems to be particularly  complex. Finally, the authors counted transcripts in zebrafish and humans that would generate non-functional proteins, but note that observed differences are very likely to be due to insufficient data.

The only new information in this review are the counts of alternative Hox transcripts in fish compared to mammals (Figure 1), but the data need to be processed further: (1) A statistical analysis that provides at least an idea of the most likely significant deviations in transcript types from the average expectation should have been performed as the basis for further ideas. (2) The discussion should strictly be done in a phylogenetic framework, where, for example, a significant trend in humans would also be verified for Bonobos.

The authors apparently used public data to compare numbers of known alternative mRNA species for all Hox genes in fish and mammal species (lines 193-194), yet a list of accession numbers for all transcripts described in the manuscripts is missing. Further questions arise, i.e. which databases where searched and is the transcript set likely to be complete? What is known about alternative transcript abundance? The authors do not provide information where (mRNA detected in particular organs) and when (age of the embryo/larva/adult) the alternative transcripts were found, yet this kind of data could corroborate or weaken the interpretations.

I think the manuscript promotes an interesting idea, but in the end I find it rather dissatisfying, as there is currently too little information what the alternative transcripts might be doing – if they are doing anything at all. The authors suggest regulatory roles in development, but their arguments of increased complexity of body regions where Hox genes are active with above average transcript variety are anecdotal and superficial (summarized for fishes in lines 242-245, 250-253 and 274-276 and for mammals in lines 335-337 and 353-357). I agree that the head-trunk boundary probably requires complex Hox gene regulations, but it is equally possible that the increase in alternative transcripts is more of a side-product, a fallout or white noise of regulatory interactions, with no true functions for these transcripts. This should be tested

Another topic are the transcripts encoding non-functional Hox variants. Here the numbers of different ranscript species are exaggerated for Hox3 and 4 in zebrafish and HOX3 plus several others in humans. However, the authors suggest themselves that available data on transcript variants may be insufficient for several species of fish (lines 369 and 375). This is likely to be true, therefore I doubt that any conclusions should be drawn from this data set.

Minor criticism:

Line 99: illogical sentence structure, as duplications result primarily in an increase of genes and only secondarily in gene losses.

Author Response

Dear Reviewer,

Thank you, for your input!

We addressed your criticism and suggestions:

Statistical testing for deviation of transcript types from expected distribution requires a sample size drastically larger, than available data, otherwise such such testing would be unreliable or even misleading. To properly address this a dedicated sets of quantatice transcriptomic data for HOX gene transcripts is needed. We expanded conclusion of our manuscript to explicitly state, that our suggestions on alternative and non-coding transcript role are purely theoretical, and just highlight a potential point of interest for an experimental investigation on the matter.

We expanded our manuscript to better reflect phylogenetic relations between mammals as well as fishes, and discuss observed differences and similarities of mammalian lineages.

We added numbers for ensembl transcriptome assemblies we used in Table 2.

Discussion on potential role of alternative transcripts was expanded to highlight that potential regulatory role is hypothetical and other viable explanations are possible. However, a dedicated test for alternative transcript role requires a separate study, including laboratory experiments, and can not be done in ten days allocated for the revision. We added a passage on necessity of such study to our manuscript conclusion.

We added discussion on insufficiency of data on non-functional transcripts and need for further study of the subject.

Explicit mention of the fact, that gene loss is secondary to duplication was added.

Round 2

Reviewer 2 Report

My main points were

-       Statistics on transcript number expectations vs. observed: I am satisfied with the author’s response.

-       Discussion in phylogenetic framework - ?

-       Databases searched, transcript ID’s: Databases searched is OK, but transcript accession numbers also need to be provided in Supplemental table, as to guide further studies (as suggested by the authors in lines 455-456)

English language:

I highly recommend that the manuscript is revised for grammar by someone with a good grasp of English.

Some of several more mistakes to correct are:

183 “case of secondary for elevated” – incomprehensible grammar

204 quantities

286 paralogous clusters

304 Tetraodon nigroviridis

188, 316, 359 kyrillic font

321 its

375 last sentence is missing a verb.

Author Response

Thank you for your prompt review.

We modified our manuscript according to your suggestions:

A separate paragraph on phylogenetic relationships and their putative impact on discussed issues was added to summarise and augment existing mentions of the issue.

We added a supplimentary table with explicit data.

We thoroughly checked our manuscript and corrected grammar issues, both listed in review and other.